# Properties of Cobalt- and Nickel-Doped Zif-8 Framework Materials and Their Application in Heavy-Metal Removal from Wastewater

**DOI:** 10.3390/nano10091636

**Published:** 2020-08-20

**Authors:** Bowen Shen, Bixuan Wang, Liying Zhu, Ling Jiang

**Affiliations:** 1School of Chemistry and Molecular Engineering, Nanjing Tech University, Nanjing 210009, China; 201861205103@njtech.edu.cn (B.S.); 201961105021@njtech.edu.cn (B.W.); 2College of Food Science and Light Industry, Nanjing Tech University, Nanjing 210009, China; jiangling@njtech.edu.cn

**Keywords:** zeolite imidazole framework-8, doped materials, petal-like structure, adsorption, heavy-metal removal

## Abstract

Heterometallic zeolite imidazole framework materials (ZIF) exhibit highly attractive properties and have drawn increased attention. In this study, a petal-like zinc based ZIF-8 crystal and materials doped with cobalt and nickel ions were efficiently prepared in an aqueous solution at room temperature. It was observed that doped cobalt and nickel had obviously different effects on the morphology of ZIF-8. Cobalt ions were beneficial for the formation of ZIF-8, while addition of nickel ions tended to destroy the original configuration. Then we compared the absorption ability for metal ions between petal-like ZIF-8 and its doped derivatives with anion dichromate ions (Cr_2_O_7_^2−^) and cation copper ions (Cu^2+^) as the absorbates. Results indicated that saturated adsorption capacities of Co@ZIF-8 and Ni@ZIF-8 for Cr_2_O_7_^2−^ reach 43.00 and 51.60 mg/g, while they are 1191.67 and 1066.67 mg/g for Cu^2+^, respectively, which are much higher than the original ZIF-8 materials. Furthermore, both the heterometallic ZIF-8 materials show fast adsorption kinetics to reach adsorption equilibrium. Therefore, petal-like ZIF-8 with doped ions can be produced through a facile method and can be an excellent candidate for further applications in heavy-metal treatment.

## 1. Introduction

Zeolite imidazole framework materials (ZIFs) constitute a subfamily of metal-organic frameworks (MOF) materials synthesized from metal ions and imidazole groups [1,2,3,4]. ZIFs have attracted a great deal of research interest because of their proper porosity, controllable morphology, and excellent chemical and thermal stability [5,6,7,8,9,10]. ZIF-8 is a typical ZIF material that is widely used in adsorption and separation of substances [11,12,13], electrochemical sensors [14,15,16], catalysis [17,18], bacteriostasis [19], and enzymes immobilization [20,21]. In recent years, ZIF-8 has been reported to have splendid performance in environmental pollution treatments. As a member of the MOFs family, ZIF materials have been proved not only to have excellent hydrothermal and chemical stability in organic solvents and alkaline aqueous solutions, but also to have the characteristics of porous and high specific surface area, which suggests that ZIF is a promising candidate for wastewater treatment [22]. Huang et al. [22] succeeded in removing 99.4% Pb^2+^ when they used excessive ZIF-8 to treat wastewater. Zhao et al. [23] found that ZIF-8 could be a rapid absorbent for Cu^2+^. Apart from metal ions removal, ZIF-8 can also adsorb some antibiotics [24] and degrade several dyes under visible light [25,26].

The morphology of the material plays a crucial role in the performance of the material. ZIF-8 in different forms of crystals will have obviously different catalytic behaviors [27,28]. For example, Zheng et al. [29] synthesized different forms of ZIF-8, which varied greatly in catalysis, gas sensing and storage. In addition, the crystal size and pores were found to have a significant influence on the absorption properties and enzyme immobilization efficiency of ZIF-8 [30,31]. Therefore, changing the shape, size and pores of the material is an important way for us to enhance its performance. There are mainly three methods to achieve this goal, including adding additives, altering the ratio of zinc to imidazole and changing synthesis conditions [1,29,32,33]. Among the above methods, addition of extra metal ions could be an efficient and simple way to improve the properties of ZIF-8. For instance, doping Fe-C groups in ZIF-8 could greatly enhance its oxygen reduction reaction (ORR) activity [34]. The doping ions are not limited to improvement in the electrochemical field, but also could enhance the absorption capability and photocatalytic ability of ZIF-8 [35,36]. Butova et al., found that addition of cobalt ions in ZIF-8 could improve its absorption for iodine molecules [35].

ZIF-8 is a network compound formed by zinc ions and 2-methylimidazole (Hmim) through coordination bonds. Nickel and cobalt are adjacent elements of zinc in the same period, and there are structural similarities in the electron arrangement outside the nucleus among them. The literature contains relatively few reports on heterometallic ZIFs and actually there is none referring to the effect of nickel dopant on ZIF-8 nanoparticle properties. However, a closer examination of available results provides a hint that the morphology and structure of zinc related MOFs could be changed by doping with nickel and cobalt ions [37,38]. Petal-like ZIF-8 is a kind of mesoporous MOFs which was previously found to be used in enzyme immobilization [39].

In this paper, we synthesized ZIF-8 with a petal shape, and reported the formation of the doped derivatives with cobalt and nickel ions. We further explored the properties of the doping materials by using a series of characterization techniques, such as X-ray diffraction (XRD), field-emission scanning electron microscopy (FESEM/EDS), nitrogen adsorption surface meter, X-ray photoelectron spectroscopy (XPS), and so on. The cobalt- and nickel-doped ZIF-8 were then used as adsorbents to investigate their performance in removing anion dichromate ions (Cr_2_O_7_^2−^) and cation copper ions (Cu^2+^) from wastewater. Their adsorption kinetics mechanism was also explored.

## 2. Experimental Section

### 2.1. Materials

All chemicals used in the study are commercially available and can be used directly without further purification. Zn(NO)_2_·6H_2_O, Cu(CH_3_COO)_2_·H_2_O, K_2_Cr_2_O_7_, Co(NO)_2_·6H_2_O and NiCl_2_·6H_2_O were purchased from Sigma (Shanghai, China). Hmim and diphenylcarbazide were purchased from Bidepharm (Shanghai, China). The concentrations of zinc nitrate, cobalt nitrate, nickel chloride aqueous solutions were adjusted to 0.5 M, respectively, while the concentration of Hmim aqueous solution was set at 1.0 M. Copper standard solution was prepared by dissolving 3.142 g copper acetate of analytical grade in distilled water to obtain 1000 mg/L solution (0.0154 M). Chromium standard solution was prepared by dissolving 0.2829 g potassium dichromate in distilled water to obtain a 100 mg/L chromium solution (0.001 M). For the preparation of diphenylcarbazide solution (2 mg/mL), dissolve 200 mg of diphenylcarbazide in 50 mL of hot ethanol. When the diphenylcarbazide is completely dissolved, dilute with distilled water to 100 mL.

### 2.2. Synthesis of Zeolite Imidazole Framework (ZIF-8), Co@ZIF-8 and Ni@ZIF-8

To synthesize ZIF-8 in an aqueous solution, 0.5 M zinc nitrate solution (10 mL) was slowly added to the aqueous solution of Hmim (1.0 M, 50 mL) in a ratio of 1:5 under constant stirring. The reaction was performed on a magnetic stirrer at room temperature for 12 h. The product was collected by centrifugation at 8000 rpm for 10 min, and washed 3 times with water. Then the obtained product was dried at 80 °C to get a dried ZIF-8 sample. Co@ZIF-8 and Ni@ZIF-8 were produced with similar method, except that Co^2+^ or Ni^2+^ solution and zinc nitrate were mixed at a ratio of 1:5 before adding to the Hmim solution.

### 2.3. Metal Ion Adsorption Experiments

Standard chromium storage solution was diluted to 30 mg/L. Then 30 mg of adsorbent was added into 50 mL chromium solution, and kept stirred at room temperature. The suspension was taken out every half an hour and diphenylcarbazide was used as a developer to measure its absorbance. In an acidic environment, Cr (VI) reacts with diphenylcarbazide to form a purple-red complex. We mixed the test solution (1 mL) with diphenylcarbazide (2 mg/mL) and diluted to 5 mL with water, and adjusted the pH to 4. The absorption wavelength can be measured at 540 nm. In the case of the copper absorption test, 50 mL of copper standard solution was added to 30 mg of adsorbent. Moreover, the reaction temperature was 60 °C, the time interval for each measurement was 30 s, and the measurement was conducted at 750 nm. Both the adsorption and photocatalytic absorbance in this experiment were obtained by the ultraviolet–visible (UV/vis) spectroscopy technique (PerkinElmer Lambda 25, Waltham, MA, USA). The test solution was placed in a glass cuvette, and a reference solution was selected for measurement.

### 2.4. Characterization of Apparatus

X-ray diffraction (XRD). The dried products were set on a quartz slide for XRD analysis (Rigaku Miniflex 600, Shangahi, China). The 2θ scan range was 5–70°, the step size was 0.008 (2θ), and the scan speed was 3° (2θ)/min.

Field-emission scanning electron microscope/energy-dispersive X-ray spectroscopy (FESEM/EDS). Observation of sizes and morphologies of ZIFs was performed using a scanning electron microscope (SEM, S-4800, Hitachi High-Technologies corporation, Tokyo, Japan). Observation samples were prepared on a copper mesh and treated with spray gold.

Nitrogen adsorption surface meter. A nitrogen adsorption surface meter (Autosorb-iQ, Quantachrome Instruments, Delray Beach, FL, USA) was used to measure the adsorption and desorption curve of the materials for N_2_. The adsorption and desorption time was 8 h. The Langmuir equation was listed based on the measurement results.

X-ray photoelectron spectroscopy (XPS). This experiment used an X-ray photoelectron spectrometer (Thermo Fisher Scientific K-Alpha, Waltham, MA, USA) for testing. The vacuum of the analysis chamber was 5 × l0^−10^ Pa, the excitation source was Alka rays (hν = 1253.6 eV), the working voltage was 15 kV, the filament current was 10 mA, and the signal was accumulated for 5–10 cycles. Test passing energy was 50 eV, step length was 0.05 eV, and C_1s_ = 284.80 eV binding energy was used as energy standard for charging correction.

### 2.5. Mathematical Equations and Methods

In this study, we used chemical kinetics and Weber–Morris equation to analyze the adsorption results. The pseudo first-order equation is of the following form [40]:(1)$$ln\left(q_{e}-q_{t}\right)=lnq_{e}-\frac{k_{1}}{2.303}t$$
$\ln\left(q_{e}-q_{t}\right)$ is plotted against the mixing time t to get a fitted straight line.

The pseudo-secondary equation is represented by the following Equation:(2)$$\frac{t}{q_{t}}=\frac{1}{k_{2}q_{e}^{2}}+\frac{t}{q_{e}}$$
$\frac{q_{t}}{q_{e}-q_{t}}$ is plotted against the mixing time t to obtain a fitted straight line.

Among them, *q_e_* is the adsorption capacity at equilibrium time, and *q_t_* is the adsorption capacity at mixing time *t*. *k*_1_ (min^−1^) represents the pseudo first-order rate constant, and *k*_2_ (g/mg/min) represents.

Weber–Morris intra-particle diffusion is derived from Fick’s second diffusion law, and its equation is expressed as:(3)$$q_{t}=k_{p,i}^{0.5}+C$$

Among them, *k_p,i_* represents the diffusion rate constant at different stages, *i* represents different stages. *C* is the intercept of the figure on the ordinate, which is only related to the thickness of the boundary layer and the degree of external mass transfer during adsorption.

## 3. Results and Discussion

### 3.1. Characterization of Petal-Like ZIF-8 and Its Derivatives

#### 3.1.1. The X-ray Diffraction (XRD) Pattern and Scanning Electron Microscope (SEM) Pictures of Petal-Like ZIF-8 and Its Derivatives

The X-ray diffraction patterns of petal-like ZIF-8 crystal and its doped derivatives are shown in the Figure 1. This XRD pattern of ZIF-8 was similar to that of the reported petal-like ZIF-8. [39] However, significant differences in 2θ shifts and peak value were observed from the doped materials. This might be attributed to the fact that the replacement of Zn^2+^ by Co^2+^ or Ni^2+^ changed the crystal structure of ZIF-8. The SEM pictures confirmed the speculation. As is shown in Figure 2, the dried ZIF-8 materials had a petal shape with smooth surface, while Co@ZIF-8 was close to a spherical particle with rougher surface and smaller size. It may be deduced that the existence of Co^2+^ can promote the formation of the shape of the ZIF-8 crystal. Saliba et al. [36] investigated the effects of cobalt ions addition on the structure of rhombic dodecahedral shaped-ZIF-8. They also found that the mixed Co@ZIF-8 crystals were smaller than ZIF-8. As also can be seen in Figure 2C,F, Ni^2+^ addition dramatically changed the petal-like ZIF-8 and made it have a layered topology. This was also exactly the converse phenomenon found by Yang et al. [41] when Ni-MOF was doped with Zn^2+^. The reason of the different behaviors of doped Co^2+^ and Ni^2+^ may lies in the fact that electronegativity of Co^2+^ is weaker than Ni^2+^, thus having more stable structures in the ZIF-8 frameworks.

#### 3.1.2. The Energy-Dispersive X-ray Spectroscopy (EDS) Mapping and X-ray Photoelectron Spectroscopy (XPS) of Petal-Like ZIF-8 and Its Derivatives

In order to understand the interaction among Zn^2+^, the doped ions and Hmim, EDS and XPS were used to analyze the structure of the materials. As shown in Figure 3, both Co^2+^ and Ni^2+^ entered the interior of ZIF-8. Nevertheless, the density of Ni^2+^ (6.04%) was much lower than that of Co^2+^ (18.09%), indicating only a small amount of Ni ions could take part in the formation of the crystals. From the XPS results, the nitrogen-containing groups of Hmim were combined (Figure 4A), as well as oxygen of the solvent. The binding energy of the Co or Ni elements in doped ZIF-8 is shown in Figure 4B. The red line, black line and blue line represented the original data, the strength of the binding energy, and fitted line, respectively. The background line fitted well with the red line, meaning that our fitting result is correct. The fitting results showed that the dopant ions existed in the material in the valence states of +2 and +3 [42,43]. In other words, the dopant ions formed coordination bonds as coordination centers in many different ways. For these materials, the main binding sites were C=N– and C–NH– in the Hmim [44]. It can be seen from Figure 4C that the binding energy and strength of C=N– and C–NH– in the doped material have been changed, which can be seen more clear in Table 1. This is because the addition of doped ions changed the previous bonding method, and the strength of the NO group binding energy of the doped material was increased. It is speculated that more NO groups were formed before the metal ions were bound. The binding energy of the original coordination center Zn ions did not have apparent change (Figure 4D, Table 1). It can be explained that the bonding mode of Zn ions was the same as before. However, the strength of the binding energy was reduced. It may be the newly added ions that competed with Zn ions to form a bond, resulting in a decrease in the strength of the binding energy. XPS results show that cobalt and nickel ions compete with zinc ions to form bonds in a coordinated manner [45].

#### 3.1.3. The Nitrogen Adsorption Experiment Analysis of Petal-Like ZIF-8 and Its Derivatives

The adsorption and desorption curves of ZIF-8 and its derivatives are shown in Figure 5A, and the shape of the curves show typical mesoporous adsorption. There is also a hysteresis loop in the curve, which shows that the pore size distribution of the material is relatively uniform. The pore size distributions of the three materials were shown in Figure 5B, and the calculation results were obtained by the Barrett–Joyner–Halenda (BJH) calculation method. The three materials all show obvious large pore diameters, with pore diameters >2 nm as the distribution of mesopores. Compared with the 3.4 Å of the general cube-shaped ZIF-8 [1], the material synthesized here has a greater advantage in pore size. The average pore diameter and surface area of the three materials were shown in Table 2. Petal-like ZIF-8 have an average pore diameter of about 8 nm. Doping with Co ions significantly increased the specific surface area of ZIF-8, but on the contrary, the average pore diameter decreased to 5 nm. Unlike the Co@ZIF-8, doping with Ni ions significantly reduced the specific surface area of ZIF-8, but the pore size was clearly increased. Expanded specific surface area and reduced pore size can influence the adsorption efficiency to heavy metals, which will be conducted in the following series of experiments.

### 3.2. Metal Adsorption Tests

#### 3.2.1. Adsorption Results of Metal Ions

ZIF-8 has zinc ion active sites and nitrogen atom active sites on imidazole. The active sites on the surface can physically and chemically adsorb anions and cations through electrostatic interaction. Various types of ZIF-8 have been used as absorbents for treatment of heavy metals. Huang et al. [22] used flaky hexahedron ZIF-8 to adsorb Pb^2+^ and Cu^2+^ with satisfactory results. However, petal-like ZIF-8 have never been used for this area. In this study, we explored the potential of the petal-shaped ZIF-8 crystal and its doped derivatives in heavy-metal treatment. Two typical metal ions Cr_2_O_7_^2−^ and Cu^2+^ were chosen to test the adsorption performance of ZIF-8 before and after doping ions. The adsorption results are shown in Table 3 and Figure 6. From the data in the table, saturated absorption capabilities of pure ZIF-8 were 25.67 mg/g for Cr_2_O_7_^2−^ and 575.00 mg/g for Cu^2+^, respectively. As for Co@ZIF-8, the absorption capabilities increased by 68% for Cr_2_O_7_^2−^ and 107% for Cu^2+^. Surprisingly, nickel ZIF-8 was able to adsorb 51.60 mg/g Cr_2_O_7_^2−^ and 1066.67 mg/g Cu^2+^. It can be seen that the adsorption capacity of ZIF-8 doped with Co and Ni had a considerable increase. That means that doping ions make the ZIF-8 surface have more active sites on surface.

#### 3.2.2. The Effect of pH on Adsorption Capacity

In order to test the effect of pH on the adsorption capacity, the adsorption capacity of ZIF-8 and its derivatives to anions and cations was tested under different pH conditions. Figure 7A shows the adsorption of chromium ions by the three materials. It can be seen that the adsorption capacity reaches the maximum value at pH 4. As the pH increases, the adsorption capacity of the material gradually decreases. This is because Cr (VI) mainly exists in the form of HCrO_4_^−^ anion under acidic conditions [46]. Therefore, the acidic conditions are beneficial to the adsorption of Cr (VI). However, when pH at 3, ZIF-8 and its derivatives will be acid decomposed, where metal ions cannot be adsorbed. Figure 7B shows the adsorption of copper ions at different pH. The maximum value is reached at pH 6. As the solution environment becomes alkaline, Cu(OH)_2_ precipitation will form, which is not conducive to the adsorption of metals ions.

#### 3.2.3. Kinetic Analysis of Adsorption Process

In order to further analyze the adsorption process of the three materials, pseudo-first-order and pseudo-second-order equations were used to model the adsorption process. As is shown in Figure 8, the data of ZIF-8 adsorption of chromium were basically distributed on a straight line in the pseudo first-order model (Figure 8A), which is simulated by Equation (1). In contrast, when using Equation (2) for the simulation, the data of ZIF-8 adsorption of chromium was not well distributed in the pseudo-second equation on a straight line, and the correlation coefficient was less than 0.8 (Figure 8B). The data of material adsorption of copper was in accordance with the pseudo-second equation (Figure 8C,D). When it came to the kinetic simulations of the adsorption results of Co@ ZIF-8 and Ni@ZIF-8, they were similar to the analysis results of ZIF-8. That meant the adsorption results of anionic chromium in ZIF-8 and doped materials were more in line with the pseudo-kinetics. The first-order model, and the adsorption of oppositely charged copper ions fitted the second-order kinetic model. The reason for the above difference is that the adsorption of chromium ions is mainly through physical adsorption, while the adsorption of copper ions is mainly by chemical adsorption.

#### 3.2.4. Diffusion Analysis in the Adsorption Process

The adsorption of porous materials can generally be divided into three stages [47]. The first stage is the diffusion of the adsorbed substance to the surface of the adsorbent, so this stage is also called the boundary layer diffusion. The second stage is a process of reducing the concentration of the adsorbed substance. The third stage is to reach an adsorption equilibrium stage, which includes the internal pore diffusion of porous materials. Therefore, here we used the Weber–Morris intra-particle diffusion equation [48] to study the adsorption process of the pores in the adsorption process.

The W–M equation shown in the figure reflects the three-stage adsorption process, which is simulated by Equation (3). The first curve reflects the process of surface adsorption, the second curve reflects the process of adsorption in the channel, and the third curve reflects the adsorption reaching dynamic equilibrium. In the chromium adsorption process, the surface adsorption amount of ZIF-8 was very small, and the adsorption in the pores accounted for a large part, about 80%, of the entire adsorption amount. But in doped materials, surface adsorption was more intense, accounting for 50% of the total adsorption (Figure 9A–C). Therefore, the improvement of surface-active groups was the key factor for increasing the amount of chromium adsorbed by doped materials. But in the process of copper adsorption, surface adsorption dominates the course (Figure 9D–F). The reason was that copper ions react with N active groups on the surface, so Co@ZIF-8 with the larger specific surface area has advantages in adsorbing copper ions. The experimental results show that the petal-shaped ZIF-8 has greater potential in adsorption capacity than the square ZIF-8, and the adsorption capacity could be further improved with doping ions [49,50]. In brief, we proved that the ion-doped ZIF-8 has an improved adsorption capacity for metal ions. The reasons for the increase in adsorption capacity include: (1) the doped ions change the morphology and pore structure of ZIF-8, which increases the adsorption capacity of negatively charged ions; (2) the doped ions increase the activity on the ZIF-8 surface, where the site increases its adsorption of positively charged ions [21,36].

#### 3.2.5. The Shape of ZIF-8 and Its Derivatives after Adsorption

Characterizations of ZIF-8 and its derivatives after adsorption of metal ions are shown in Figure 10 and Figure 11, respectively. Figure 10 displayed that the shape of the materials was in disorder, and chromium ions have caused devastating damage to the materials during the adsorption process. In addition, after adsorption tests, the proportion of chromium in Co@ZIF-8 (16.60%) and Ni@ZIF-8 (12.65%) was much larger than the proportion of chromium in ZIF-8 (3.58%). The reason may lie in the fact that doping ions addition weakened the bond between zinc ions and imidazole groups, thus making invasion of chromium ions more easily. For doped derivatives, the ratio of Co/Zn decreased from 23.08% to 12.17%, and nickel ions were almost completely lost. Obviously, part of the cobalt ions and zinc ions and almost all the nickel ions were substituted by chromium ions.

Different from chromium ions adsorption results, the adsorption of copper ions made all the materials become elliptical shaped (Figure 11), which was similar to Cu-imidazole MOF [51]. Moreover, almost no metal ions were left behind but copper ions. That means that all the metal ions in ZIF-8 and its derivatives were substituted by copper ions. Copper ions and imidazole groups form bonds during adsorption, which might explain why ZIF-8 has a large adsorption capacity for copper ions.

Figure 12 also presents the XRD patterns of ZIF-8 and its doped derivatives before and after the adsorption of ions. It can be seen that the three materials are all highly crystalline and the XRD patterns are in good agreement with the previous report [39]. It is also worth noting that the XRD pattern after the adsorption of chromium ions is nearly the same as the original one, although more derivative peaks appear after the adsorption. However, all three XRD patterns involving in the adsorption of copper are slightly lower than those of virgin ZIF-8 and chromium-adsorbed ones.

## 4. Conclusions

In conclusion, we employed cobalt ions and nickel ions in the synthesis of petal-like ZIF-8, and obtained the new doped materials Co@ZIF-8 and Ni@ZIF-8. Through a series of characterizations, it could be observed that the doped materials had a different form and a different surface from the original material. They could have a larger adsorption capacity as absorbent for both anion dichromate ions and cation copper ions. The two adsorbents of Co@ZIF-8 and Ni@ZIF-8 showed fast adsorption kinetics and were more accurately described by the pseudo-second order model. In addition, doping ions in ZIF-8 could change its morphology and make it have more active sites. Therefore, petal-like ZIF-8 could have great potential in the application of heavy-metal absorption from wastewater after doping with Co or Ni ions.

## Figures and Tables

**Figure 1 nanomaterials-10-01636-f001:**
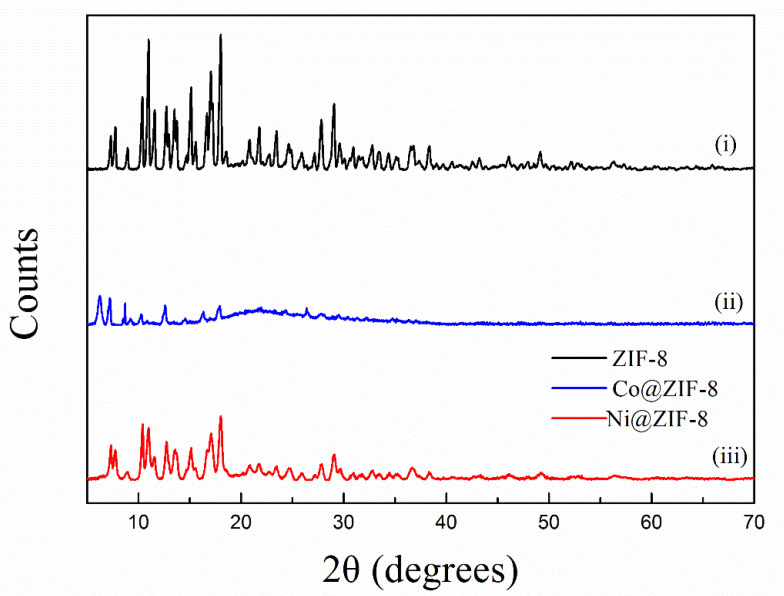
X-ray diffraction (XRD) pattern of ZIF-8 and its derivatives.

**Figure 2 nanomaterials-10-01636-f002:**
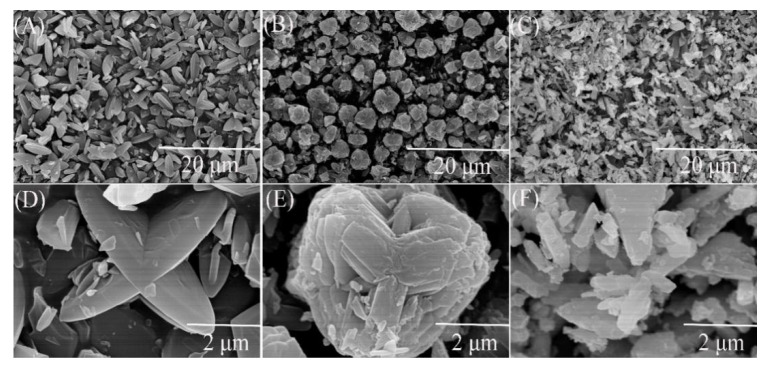
Scanning electron microscope (SEM) images of ZIF-8 and its derivatives. (**A**) ZIF-8 in scale of 20 μm; (**B**) Co@ZIF-8 in scale of 20 μm; (**C**) Ni@ZIF-8 in scale of 20 μm; (**D**) ZIF-8 in scale of 2 μm; (**E**) Co@ZIF-8 in scale of 2 μm; (**F**) Ni@ZIF-8 in scale of 2 μm.

**Figure 3 nanomaterials-10-01636-f003:**
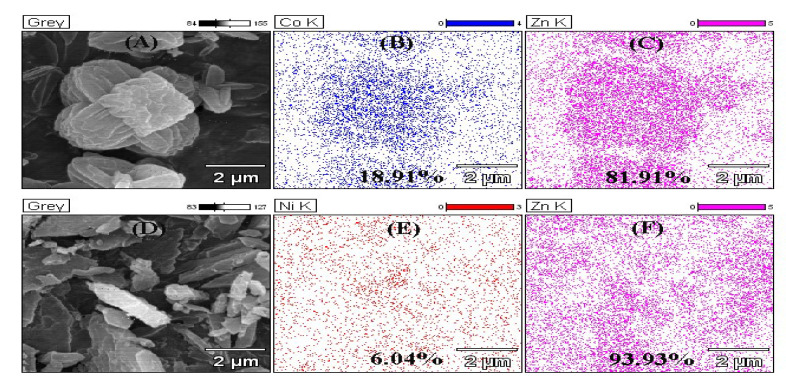
Energy-dispersive X-ray spectroscopy (EDS) mapping of ZIF-8 and its derivatives. (**A**) SEM of Co@ZIF-8; (**B**) Distribution of Co in Co@ZIF-8; (**C**) Distribution of Zn in Co@ZIF-8; (**D**) SEM of Ni@ZIF-8; (**E**) Distribution of Ni in Ni@ZIF-8; (**F**) Distribution of Zn in Ni@ZIF-8.

**Figure 4 nanomaterials-10-01636-f004:**
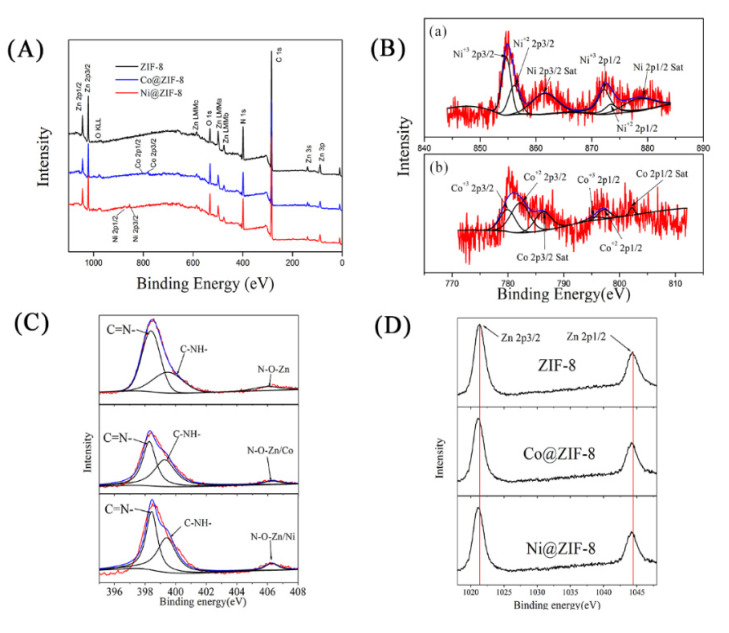
(**A**) X-ray photoelectron spectroscopy (XPS) wide scan spectra of ZIF-8 materials and doped materials. (**B**(a)) Ni 2p, (**B**(b)) Co 2p, (**C**) N 1s, (**D**) Zn 2p.

**Figure 5 nanomaterials-10-01636-f005:**
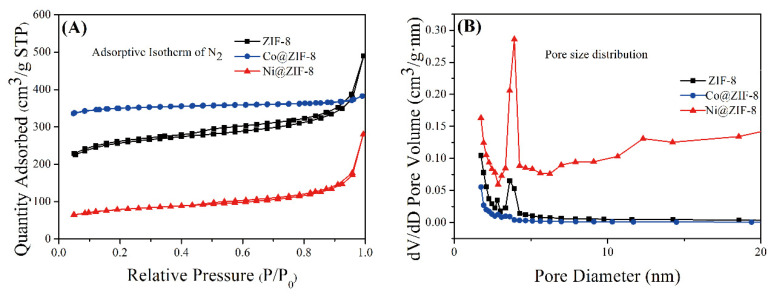
(**A**) Adsorptive Isotherm (N_2_) of ZIF-8 and its derivatives; (**B**) pore size distribution of ZIF-8 and its derivatives.

**Figure 6 nanomaterials-10-01636-f006:**
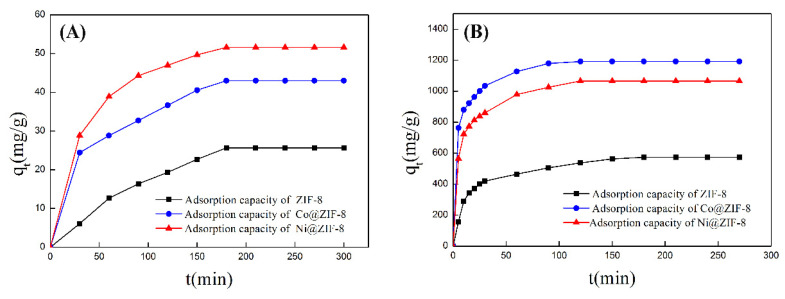
(**A**) Adsorption curve of ZIF-8 and doping materials for chromium ions; (**B**) adsorption curve of ZIF-8 and doping materials for copper ions.

**Figure 7 nanomaterials-10-01636-f007:**
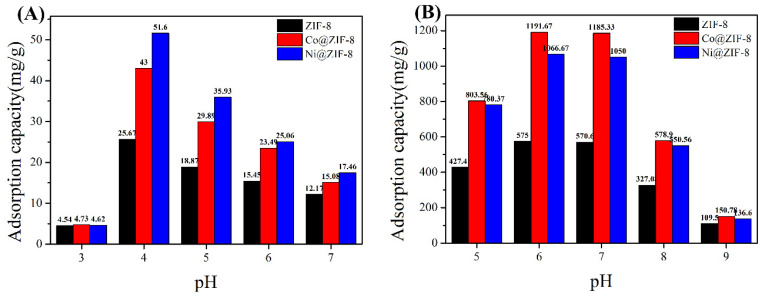
(**A**) The influence of pH on the adsorption capacity of chromium ion; (**B**) the influence of pH on the adsorption capacity of copper ion.

**Figure 8 nanomaterials-10-01636-f008:**
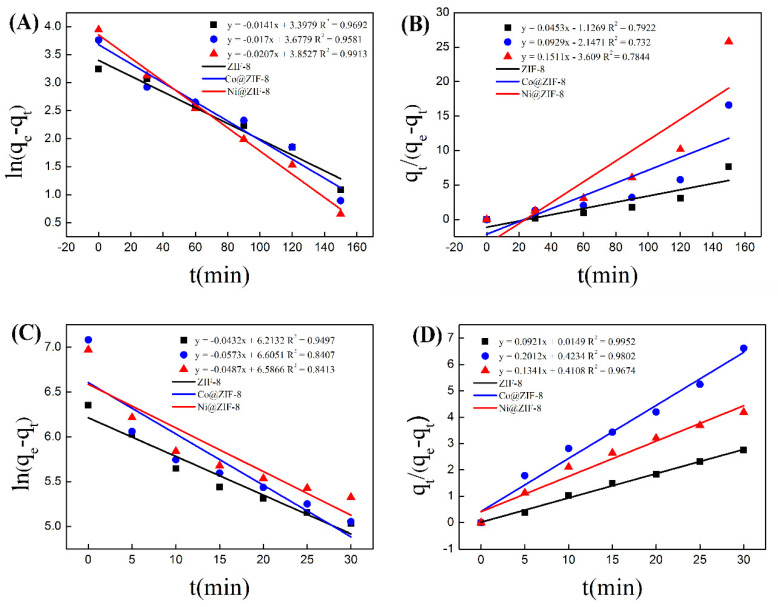
(**A**) Standard curve of pseudo first order kinetics for chromium adsorption; (**B**) standard curve of pseudo second order kinetics for chromium adsorption; (**C**) standard curve of pseudo first order kinetics for copper adsorption; (**D**) standard curve of pseudo second order kinetics of copper adsorption.

**Figure 9 nanomaterials-10-01636-f009:**
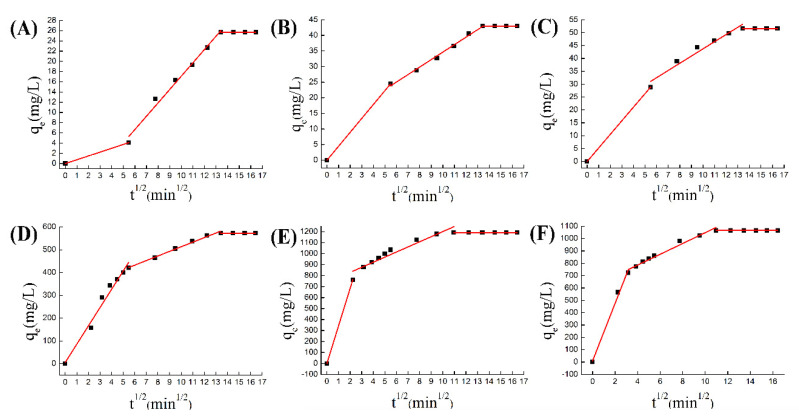
(**A**) W–M equation for ZIF-8 adsorption of chromium; (**B**) W–M equation for Co@ZIF-8 adsorption of chromium; (**C**) W–M equation for Ni@ZIF-8 adsorption of chromium; (**D**) W–M equation of ZIF-8 adsorption of copper; (**E**) W–M equation of Co@ZIF-8 adsorption of copper; (**F**) W–M equation of Ni@ZIF-8 adsorption of copper.

**Figure 10 nanomaterials-10-01636-f010:**
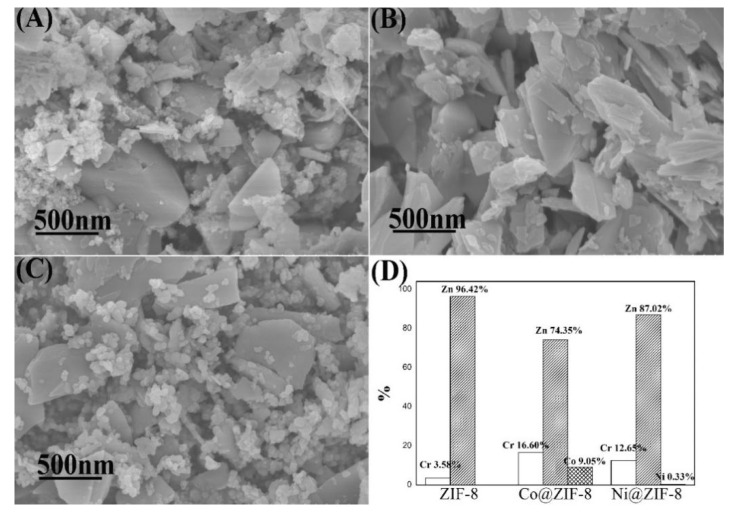
Characterization ZIF-8 and its derivatives after adsorption of chromium ions. (**A**) SEM of ZIF-8; (**B**) SEM of Co@ZIF-8; (**C**) SEM of Ni@ZIF-8; (**D**) proportion of metal elements in the material after adsorption of chromium ions.

**Figure 11 nanomaterials-10-01636-f011:**
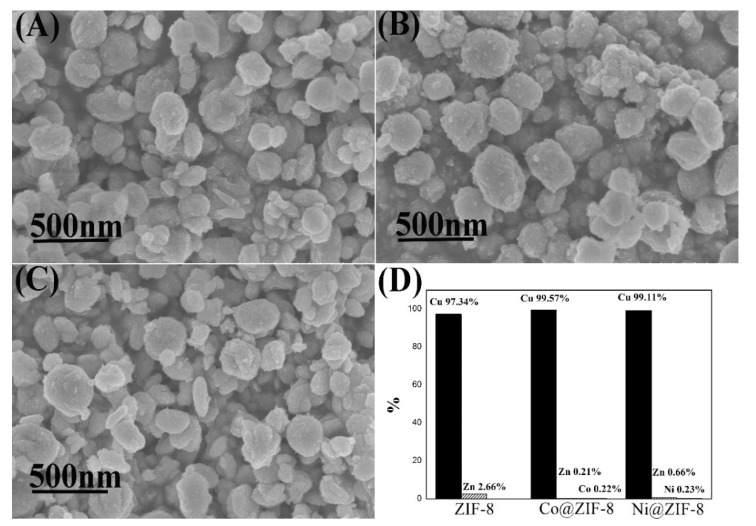
Characterization ZIF-8 and its derivatives after adsorption of copper ions. (**A**) SEM of ZIF-8; (**B**) SEM of Co@ZIF-8; (**C**) SEM of Ni@ZIF-8; (**D**) proportion of metal elements in the material after adsorption of copper ions.

**Figure 12 nanomaterials-10-01636-f012:**
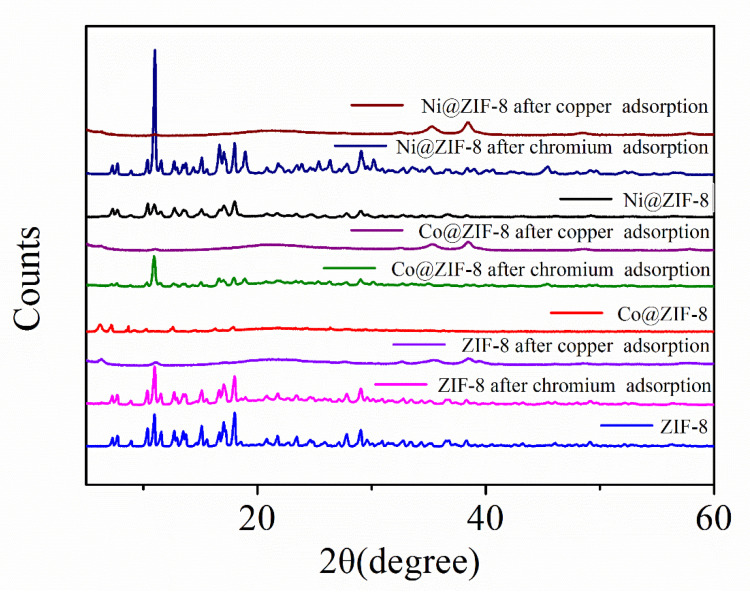
The XRD patterns of ZIF-8 and its derivatives after adsorption.

**Table 1 nanomaterials-10-01636-t001:** Changes in binding energy of three materials.

Materials	Binding Energies of C=N–(eV)	Binding Energies of C–NH–(eV)	Binding Energies of Zn 2p3/2 (eV)	Binding Energies of Zn 2p1/2 (eV)
ZIF-8	398.7	399.7	1021.3	1044.8
Co@ZIF-8	398.2	399.1	1020.7	1044.2
Ni@ZIF-8	398.3	399.2	1020.7	1044.2

**Table 2 nanomaterials-10-01636-t002:** Aperture data of ZIF-8 and its derivatives.

Material	ZIF-8	Co@ZIF-8	Ni@ZIF-8
BET Surface Area (m^2^/g STP)	840.884	1072.73	259.93
Average pore diameter (nm)	8.111	4.7861	12.0861

**Table 3 nanomaterials-10-01636-t003:** Adsorption results of ZIF-8 and its doped derivatives.

Material	Adsorption of Cr_2_O_7_^2−^ (mg/g)	Improvement Compared to ZIF-8	Adsorption of Cu^2+^ (mg/g)	Improvement Compared to ZIF-8
ZIF-8	25.67	-	575.00	-
Co@ZIF-8	43.00	65.7%	1191.67	107.25%
Ni@ZIF-8	51.60	101.01%	1066.67	85.51%

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
