# Peer review of "Properties of Cobalt- and Nickel-Doped Zif-8 Framework Materials and Their Application in Heavy-Metal Removal from Wastewater"

_nanomaterials, 2020, doi:10.3390/nano10091636_

Round 1

Reviewer 1 Report

Introduction

1) The authors should describe the reason for the use of ZIFs as an adsorbent for the removal of heavy metals. Selectivity?? Cost??

2) Line 57. The authors should separate the sentence from the present paragraph. And please describe the outline of this manuscript.

Materials and Methods

3) Section 2.2. I cannot find the information about the use of zeolite. What kind of zeolite used in this study?? And how zeolite was reacted for the synthesis of ZIF??

4) Line 85: What is meaning of time interval?? For what??

5) Line 98: It should be moved to the end of Section 2.3.

Results and Discussions

6) Section 3.1.: The paragraphs are too long to understand. It should be separated to have each topic.

7) Please provide the specific surface area of three materials. Not given in Table 1.

8) Mathematical equations in Page 7 should be described in Materials and Methods.

9) The author should describe how the doped ZIFs improved the adsorption capacity of Cu and Cr?? What is the mechanism?

10) The authors should describe the pH performed in this experiment because the removal mechanism is dependent on pH.

11) The authors should perform the batch experiments under different pH.

Reviewer 2 Report

Author have written the manuscript on application of doped ZIF-8 materials for heavy metal removal from the waste water. The work is of significance and can add to the knowledge available in the literature. The real possible application can still not be concluded from the available results. But it is a step towards that. 

Even though the work is important, I have some concerns with the manuscript. I have listed them below.

  1. In materials section, add details of purity and supplier of the chemicals.

  2. In synthesis of ZIFs, experimental details need elaboration. What were the volumes needed for each solution? What were the ratio of Co/Ni Nitrates + Zinc nitrates with Hmim? What do you mean by “certain ratio” in line 72 on page 2.

  3. Experimental details need to elaborate on the copper absorption test as well. I do not see the procedure clearly from the experimental section.

  4. Also, what was the concentration of diphenylcarbazide in the experiment? What was the used reaction and suspension volume? These details are needed from the reproducibility perspective.

  5. In results section, authors should add simulated ZIF-8 XRD pattern in Figure 1 for clear comparison.

  6. Line 109: Can the authors elaborate on this statement :

    It may be deduced that the existence of Co2+ is beneficial to the nucleation of ZIF-8 crystals

    I would like to know how did the authors arrived at this conclusion.

  7. Also, since Co@ZIF-8 has been studied before, how does the XRD from the literature looks like for this MOF. The diffraction pattern is significantly impacted in comparison to ZIF-8.

  8. Authors should also include the high resolution C- spectra for XPS. How was the fitting done? Was there a correction for binding energy for C? Experimental details do not describe the XPS procedure. Does the C spectra shows the peaks of C-N bonds as well?

  9. Authors should add the respective changes in binding energies from XPS studies (either in text or as a table). The text on page 4 doesn’t describe the numbers. The drawn conclusions with respect to the shifts in binding energies are not clear without these numbers. Also, please explain the reason behind the shift in C-N binding energies due to presence of Ni/Co. Are these changes visible in Ni/Co XPS spectra?

  10. What were the calculated surface areas in the N2 adsorption studies? Also, add the adsorption isotherm of the study in the manuscript.

  11. Authors have described

    “In contrast, the data of ZIF-8 adsorption of copper was not well distributed in the pseudo-second equation on a straight line, and the correlation coefficient was less than 0.8.” on Page 8 line 192-194.

    However, the data in Figure 7 says differently. The R2 is greater than 0.96 and the second order seems to fit better. Also there is a typo in the legend of the figure 7(D). It says Pseudo-first order instead of second order.

  12. Authors concluded that Co@ZIF-8 has advantage in adsorption due to high specific surface area. What are the exact values? Also, there is a significant decrease in the pore volume. How does it impact the surface/volume ratio (in other words, the available adsorption sites). Without this information, it is not possible to conclude that Co@ZIF-8 has advantage. Ni@ZIF-8 shows significant improvement as well. But the data on standard deviation is missing to conclude the real impact. Authors should add it as well for the adsorption studies. It helps in getting a feel on reliability of the data and the made conclusions.

  13. How did the XRD of the materials looked like after adsorption?

  14. If there was loss of Co/Ni from the material, does it mean leaching of these ions to the solution? In such a case, the waste-water would be polluted by addition of these ions. Also, since the composition of material changes, can the material be re-used again? What would be the authors’ opinion on recycling experiments? This will really limit the real industrial application of the materials.

Round 2

Reviewer 1 Report

The authors tried their best to reflect the reviewer's comments and improve the quality of the manuscript. I belive that this manuscript has been improved to be published in this journal. 

Reviewer 2 Report

Authors have addressed the comments from both the reviewers sufficiently. Minor grammatical changes and spell-check shall be done in proof-reading stage. Other than this, it can be accepted in current form.